# The Role of Alpha-Linolenic Acid and Other Polyunsaturated Fatty Acids in Mental Health: A Narrative Review

**DOI:** 10.3390/ijms252212479

**Published:** 2024-11-20

**Authors:** Camilla Bertoni, Cecilia Pini, Alessandra Mazzocchi, Carlo Agostoni, Paolo Brambilla

**Affiliations:** 1Department of Veterinary Sciences for Health, Animal Production and Food Safety, University of Milan, 20122 Milan, Italy; camilla.bertoni@unimi.it; 2Department of Pathophysiology and Transplantation, University of Milan, 20122 Milan, Italy; cecilia.pini@unimi.it (C.P.); paolo.brambilla1@unimi.it (P.B.); 3Department of Clinical Sciences and Community Health, University of Milan, 20122 Milan, Italy; alessandra.mazzocchi@unimi.it; 4Pediatric Area, Fondazione IRCCS Ca’ Granda Ospedale Maggiore Policlinico, 20122 Milan, Italy; 5Department of Neurosciences and Mental Health, Fondazione IRCCS Ca’ Granda Ospedale Maggiore Policlinico, 20122 Milan, Italy

**Keywords:** alpha-linolenic acid, mental health, schizophrenia, depression, bipolar disorder, PUFA, psychosis, neurodevelopment, dementia, Alzheimer’s disease, ADHD

## Abstract

The present review investigates the relationship between polyunsaturated fatty acids (PUFAs) and mental health disorders, such as dementia, psychosis, schizophrenia, Alzheimer’s disease, anorexia nervosa, and impairment problems in animals and human models. Data were collected from a variety of studies: randomized intervention trials, observational and interventional studies, case reports, and epidemiological studies. The evidence suggests that PUFAs are beneficial for mental health, brain function, and behavior. ALA, EPA, and DHA have very significant neuroprotective properties, particularly in inducing changes to the synaptic membrane and modulating brain cell signaling. In the case of neurodegenerative disorders, PUFAs incorporated into cellular membranes have been shown to protect against cell atrophy and death. The formal analyses of the included studies pointed to a decrease in ALA, EPA, and DHA levels in various populations (e.g., children, adolescents, adults, and seniors) presenting with different types of mental disorders. These results indicate that PUFA supplementation may be considered as an innovative therapeutic strategy to reduce the risk of neuronal degeneration.

## 1. Introduction

Omega-3 fatty acids, particularly alpha-linolenic acid (ALA), eicosapentaenoic acid (EPA), and docosahexaenoic acid (DHA), play critical roles in mental health, neurodevelopment, brain functions, and behavior. ALA is considered an essential fatty acid because humans lack the enzymes (Δ12 and Δ15 desaturase) needed to synthesize it, meaning it must be acquired directly from dietary sources such as walnuts, chia seeds, flax seeds, and hemp seeds and oil [1,2]. Once in the body, ALA undergoes conversion to longer-chain fatty acids with 20 or 22 carbon atoms through cellular desaturation and elongation processes. Specifically, in the endoplasmic reticulum, ALA is converted to C18:4 ω-3 by the enzyme Δ6 desaturase. Next, two carbons are added by elongase, followed by desaturation at the Δ5 position by Δ5 desaturase, resulting in C20:5 ω-3 (EPA). Further elongation produces C22:5 ω-3, and finally, peroxisomal β-oxidation completes the synthesis of DHA [3]. In addition to endogenous synthesis, EPA and DHA can also be obtained directly from the diet, especially from fatty fish and fish oils. ALA is metabolized in vivo and partially converted to EPA and DHA. Alternatively, EPA and DHA can also be obtained by consuming fish oil [4]. ALA contributes to brain function and protection without adverse effects [5,6] and supports the proper development of the central nervous system, particularly in childhood [7]. Imbalances in polyunsaturated fatty acids (PUFAs) are implicated in various central nervous system diseases, including neurodegenerative disorders, due to the brain’s reliance on PUFAs like DHA, EPA, and ALA to maintain structural integrity and function [8]. Research on dietary PUFA incorporation, particularly DHA, shows that PUFAs in the brain begin accumulating in utero and continue into early childhood, maintaining high concentrations throughout life to support synaptic function and neuronal resilience [9,10]. As omega-3 and omega-6 PUFAs are integral to biological processes like metabolism, neurotransmission, synaptogenesis, and inflammation, their role in mental health throughout the lifespan continues to be a subject of intensive study [11,12,13].

Considering PUFAs from a lifespan perspective and their potential role in the development of mental disorders, it is important to note that early life is marked by an increased demand for PUFAs, particularly DHA, for optimal brain development and function. Adequate levels of DHA and arachidonic acid (AA), supplied via the mother through the amniotic fluid in utero and breast milk post-birth, are crucial for the neurological and myelination processes that are necessary for fetal and infant development [14,15]. Deficiencies in omega-3 PUFAs, particularly DHA, have been associated with neurodevelopmental disorders such as attention-deficit/hyperactivity disorder (ADHD) and autism spectrum disorder (ASD), which typically manifest in childhood and can lead to long-term cognitive and behavioral challenges [16,17]. Studies have shown that omega-3 supplementation, particularly DHA and EPA, may improve symptoms in children with ADHD, indicating the importance of PUFA balance in early neurodevelopment [18]. Young adulthood sees an increased incidence of mood disorders, such as major depressive disorder (MDD), and psychotic disorders, including schizophrenia (SCZ) [19]. Deficits in omega-3 PUFAs are commonly found in patients with these conditions, with studies showing that first-episode, drug-naïve SCZ patients have significant abnormalities in the fatty acid composition of peripheral tissues compared to controls [18].

These abnormalities are partially normalized with antipsychotic treatment, suggesting a link between PUFA levels and the onset of psychiatric symptoms. Omega-3 PUFAs, particularly EPA, are believed to modulate brain cell signaling pathways, including monoamine regulation, receptor properties, and signal transduction, all of which may contribute to mood stabilization and symptom improvement in MDD and SCZ [8,20,21]. In middle and older adulthood, the prevalence of neurodegenerative conditions, such as Alzheimer’s disease (AD) and dementia, rises significantly. DHA is a primary component of neuronal membranes in regions critical to memory and cognition, such as the hippocampus and cortex, and low levels of DHA are associated with an increased risk of cognitive decline [16,22]. Omega-3 supplementation has shown promise in delaying cognitive decline and neurodegeneration, potentially due to its anti-inflammatory and antioxidative properties, as well as its role in neurogenesis and brain-derived neurotrophic factor (BDNF) enhancement [23,24]. Furthermore, animal studies indicate that ALA influences hippocampal and cortical health by modulating synaptic transmission, inhibiting glutamatergic transmission, and activating potassium channels, which collectively support neurogenesis and neuronal preservation [9,25,26,27].

Based on these considerations, this review aims to explore the correlation between omega-3 PUFA supplementation, particularly ALA, and mental health across different stages of life. By focusing on age-specific psychiatric disorders, we seek to clarify the potential of PUFAs in preventing and managing these conditions, examining how omega-3 supplementation may impact mental health outcomes from childhood through late adulthood.

## 2. Results

In the following section, we will examine the primary clinical evidence across key mental health disorders, structured according to the likely age of onset from a lifespan perspective. This includes neurodevelopmental disorders, mood disorders, depression and psychosis, eating disorders, and cognitive disorders, with a particular emphasis on neurodegenerative disease prevention. In Table 1 we summarized all different type of studies, the enrolled population, the area of investigation, type of nutrients, and conclusions.

### 2.1. Neurodevelopmental Disorders

Neurodevelopmental disorders are a group of conditions that manifest early in development, often before school age, and impact cognitive, motor, social, and emotional functioning [28]. These disorders, which include attention-deficit/hyperactivity disorder (ADHD), autism spectrum disorder (ASD), intellectual disabilities, and learning disorders, are characterized by delays or abnormalities in brain development that affect daily functioning and adaptability. Genetic, environmental, and epigenetic factors interact in complex ways to influence neurodevelopment, with a critical period for brain growth occurring in utero and extending through early childhood [29]. The research has increasingly identified links between neurodevelopmental disorders and metabolic imbalances, including imbalances in polyunsaturated fatty acids (PUFAs), particularly omega-3 and omega-6 fatty acids, which are crucial for brain development and function [30]. This review examines the association between omega-3 intake and improvements in ADHD-related symptoms, impulsivity, and motor and psychomotor development. A prospective cohort study by Gustafsson et al. [31] showed that, in a group of 68 pregnant women, the n-6/n-3 (*p* = 0.04) and AA/EPA ratios (*p* = 0.007) were significantly higher in those experiencing heightened ADHD symptoms, while the EPA levels were lower (β = −0.24, *p* = 0.017). In contrast, no reliable differences were found in the plasma concentrations of dietary ALA. The randomized intervention, two-arm, controlled trial of Pinar-Martì, Gignac, et al. [32] showed that ALA intake can improve sustained attention (score: −11.26 ms; 95% CI = −19.92, −2.60; *p* = 0.011), fluid intelligence (score: 1.78; 95% CI = 0.90, 2.67; *p* < 0.0001), and ADHD symptoms (score: −2.18; 95% CI = −3.70, −0.67; *p* = 0.0050) in 771 adolescents with an average age of 13.9 who adhered to the walnut intake intervention (30 g/day).

The association between ALA and attention in healthy adolescents has also been investigated [33]. Interestingly, in contrast to DHA, a higher intake of ALA was not associated with better attentional performance, although a higher amount of ALA in red blood cells seemed to have a positive effect on impulsivity. Nishi et al. [34] observed a direct correlation between ALA-rich walnut consumption and improved cognitive reaction time and attentional function, while another cross-sectional study on adolescents [32] found an association between increased erythrocyte ALA levels and reduced impulsivity, a key feature of many psychiatric disorders. The results of this study support the hypothesis that ALA may have a beneficial effect on cognition.

The relationship between mean omega-3 concentrations in breast milk samples and psychomotor development in exclusively breastfed infants has been investigated [35]. In this observational study, associations were found between breast milk DHA (β = 0.275; *p* ≤ 0.05), ALA (β = 0.432; *p* ≤ 0.05), and DHA (β = 0.423; *p* ≤ 0.05) levels, and motor development. A total of 39 pregnant women were enrolled, and 39 breastfed infants were studied. No particularly significant associations were found between these nutrients and psychomotor development.

### 2.2. Eating Disorders

Nutritional and eating disorders (EDs), such as anorexia nervosa (AN) and bulimia nervosa (BN), commonly emerge during childhood or adolescence, with a traditional peak age of onset between 16 and 19 years. However, recent evidence indicates a downward shift in the age of onset, with cases now documented in children as young as 8–9 years (Italian National Institute of Health). The energy deficit in eating disorders, including anorexia nervosa (AN), results from a low intake of calories and PUFAs. The data in the literature are conflicting regarding deficiencies in essential fatty acids (EFAs), including ALA and LA [36,37]. It has been observed that the red blood cells of individuals with AN have greatly reduced levels of n-3 in the absence of overt ALA and LA deficiency. However, a negative correlation was found between fat mass and ALA in an observational study of 22 women [37]. On the other hand, Nomura et al. [36] found no statistically significant association between n-3 levels and the risk of AN. This Mendelian randomization study that included 72.517 participants does not support the hypothesis that PUFAs may reduce the risk of an eating disorder.

### 2.3. Psychosis

Psychosis is a mental health condition characterized by disruptions in perception, thought processes, and emotional responsiveness, leading to symptoms like hallucinations, delusions, and disorganized thinking. Psychotic disorders, including schizophrenia (SCZ), schizoaffective disorder, and brief psychotic disorder, have a typical onset in late adolescence or early adulthood, though symptoms can emerge across a wide age range [38]. Alterations in PUFA levels are associated with neuropsychiatric disorders, including SCZ [39]. A 12-week randomized interventional study enrolled 81 ultra-high-risk (UHR) subjects for psychosis with an average age of 16.4 years [40]. In this case, EPA and DHA from oil marine fish (220 mg of PUFAs/day) were administered for 12 weeks. The study examined the fatty acid composition of the erythrocyte membrane and found significant ALA deficits in the patients at high risk of psychosis, suggesting that PUFA supplementation may be effective in preventing psychotic onset. The concept of a PUFA imbalance in patients with psychosis was further explored by Rice et al. [41], who assessed omega-3 levels in the erythrocytes of individuals at ultra-high risk (UHR) for psychosis, comparing them with data from healthy controls. Except for DHA, significant deficits in PUFAs, including ALA, EPA, LA, and AA, were observed in the UHR patients compared to the healthy controls (specifically, ALA 0.19 vs. 0.29 (*p* < 0.001), EPA 0.48 vs. 0.57 (*p* < 0.001), LA 6.26 vs. 6.81 (*p* < 0.001), and AA 15.52 vs. 17.86 (*p* < 0.001)). These findings provide a basis for PUFA-based interventions for emerging psychosis. An observational study by Rog et al. [42] aimed to determine differences in nutritional status and PUFA metabolism in 80 patients aged 18 to 65 years with SCZ compared to controls. Although no significant differences in serum omega-3 concentrations (AA, DHA, EPA, and ALA) were found between the patients and healthy individuals, lower omega-3 levels were detected in the SCZ patients.

A two-sample Mendelian randomization study conducted by Jones et al. [43] sought to estimate the effects of omega-3 on SCZ risk. The analyses indicated that higher DHA concentrations were associated with a lower risk of SCZ (OR = 0.83; 95% CI: 0.75–0.92), while elevated ALA concentrations were linked to an increased risk (OR = 0.07; 95% CI: 0.98–1.18). Numerous studies in the literature have confirmed that schizophrenic patients have lower blood omega-3 levels than healthy individuals [19,44]. Given the mixed results, further research is necessary to clarify whether omega-3 supplementation can effectively prevent the onset of schizophrenia and psychotic symptoms.

### 2.4. Affective Disorders

Depression is a major cause of disability worldwide, affecting approximately 300 million people [45]. It is becoming increasingly common in childhood and adolescence and is a risk factor for chronic and recurrent depression in adulthood [46]. Gracious et al. [47], in a randomized, placebo-controlled trial with 51 children and adolescents with an average age of 13 years, investigated whether ALA supplementation (550 mg of ALA) with flaxseed oil for 16 weeks could safely reduce the severity of symptoms associated with bipolar disorder (BD). It was observed that overall, symptom severity was negatively correlated with the final serum omega-3 composition (ALA, EPA, and AA). Therefore, the studies reported here suggest that dietary omega-3 PUFA deficiency may contribute to the development of mood disorders and that supplementation may offer a new treatment option [48]. The fatty acid composition in the red blood cells of 95 adolescents aged 13 to 17 years old was analyzed in a case–control study [49], and lower levels of EPA and DHA were found in the cases compared to the controls (EPA: 0.41 ± 0.11 vs. 0.46 ± 0.12, *p* < 0.001; DHA: 4.07 ± 1.04 vs. 4.73 ± 1.04, *p* < 0.001). Adolescents with a higher n-3 serum concentration (4.84 ± 1.16) had lower odds of depression (OR = 0.49 [95% CI: 0.32–0.71]), whereas a high n-6/n-3 ratio (5.51 ± 1.25 vs. 4.96 ± 1.08) was associated with higher odds of depression (OR = 1.58 [95% CI: 1.14–2.25]). There are conflicting data in the literature regarding the role of omega-3 fatty acids, especially ALA, in depression. Zeng et al. [50], using a two-sample MR analysis, claimed that only adipic acid and EPA have protective effects against the risk of depression, while ALA is a potential risk factor for this disorder. In contrast, the ELSA Brazil study, a multicenter prospective cohort study [51], found that the risk of developing depressive episodes was reduced by increasing the intake of omega-3 fatty acids, with significant associations with ALA consumption (OR = 0.71: 95% CI [0.59–0.91]). A total of 13,879 adults aged 39 to 65 years old were enrolled in the trial. Regarding the risk of postpartum depression, the postpartum mental health of 250 primiparous women who consumed ALA from perilla and fish oil for 12 weeks was also studied. The association between fatty acids in maternal red blood cells and risk factors for mental health was examined, and it was found that abundant amounts of this fatty acid were present in the women’s red blood cells (OR = 0.23, 95% CI: 0.06, 0.84, *p* = 0.018), suggesting that ALA during pregnancy may stabilize mental health and reduce episodes of postpartum depression [2].

Regarding anxiety symptoms, higher intakes of oleic acid (OR 0.25; 95% CI 0.09–0.67; *p* trend = 0.002), ALA (OR 0.07; 95% CI 0.02–0.23; trend *p* < 0.001), and n-3/n-6 (OR 0.56; 95% CI 0.24–1.03; *p* trend = 0.02) were found to correlate with a lower OR for anxiety in 300 women between 18 and 49 years old, with an average age of 33.5 years [52]. 

PUFAs have also been shown to be effective in the treatment of bipolar disorder (BD), although the specific role of PUFAs in the treatment of BD remains unclear. Evans et al. [53] studied 27 bipolar patients in their cross-sectional study and found that those with a history of suicide had very high levels of AA/DHA, AA/EPA, and n-6/n-3. This suggests that these ratios may be considered markers of suicide risk.

### 2.5. Cognitive Impairment

It is estimated that over 55 million people suffer from dementia [54].

Currently, despite pharmacological progress, there are no effective therapies to treat cognitive impairment, so it is very important to find therapeutic alternatives. One example of prevention comes from adherence to a plant-based diet, such as the Mediterranean diet, which is rich in PUFAs, fiber, and polyphenols, and is associated with a lower risk of age-related cognitive decline. Data from scientific studies have shown that walnuts, which are rich in omega-3, especially ALA, LA, and oleic and palmitic acids, are effective in preventing AD [55,56]. Furthermore, other studies have revealed that these foods increase cognitive function in elderly subjects [57,58,59,60]. Studies by Esselun et al. [61,62] showed that feeding animal models food enriched with 6% walnuts led to a significant improvement in spatial memory, due to an increase in the length and number of neurites. This leads to the conclusion that the essential fatty acids contained in walnuts, such as ALA and LA, might have beneficial properties on neuronal development and Alzheimer’s disease-associated amyloid-beta in a cell model with early-stage disease. In a cross-sectional investigation study by Current et al. [63], an inverse association was found between ALA and cognitive impairment, and the opposite was found for LA intake, in 833 adults with an average age of 67.1 years. Indeed, ALA has been shown to improve learning, and semantic, spatial, and short-term memory in the elderly, thus preventing cognitive decline. The fatty acid profile of red blood cells in patients with mild cognitive impairment (MCI) and AD was determined; in the 78 patients with MCI, a significant increase in stearic acid (*p* = 0.0001), AA (*p* = 0.003), and tricosanoic acid (*p* = 0.007) levels was found. In contrast, the levels of both ALA and DHA were reduced in patients with MCI and AD (*p* = 0.0005 and *p* = 0.00003, respectively). This suggests that fatty acid testing may prove useful as potential diagnostic biomarkers reflecting an increased risk of dementia [64]. A scientific explanation for their function is that PUFAs omega-3 (ALA, EPA, and DHA) enhance neurogenesis in the adult hippocampus, promote synaptic plasticity, and modulate synaptic protein expression. This allows PUFAs, especially ALA, to be used as a therapeutic for neurodegenerative diseases [65]. The cognitive function of 60 healthy elderly subjects with an average age of 72 years was tested after using a diet enriched with 2.2 g ALA [66]. After 12 weeks of treatment, improvements in verbal fluency scores were observed compared to the control group (0.30 ± 0.53 vs. 0.03 ± 0.49, *p* < 0.05). This suggests that daily ALA supplementation may be effective in improving cognitive function despite age-related decline. Yamagishi et al. [67] performed an intra-cohort case–control study and found an inverse relationship between serum ALA and the risk of dementia impairment (OR = 0.57, 0.51 and 0.61; 95% CI) in 7586 Japanese individuals aged 40 to 74 years old. This result led to the identification of ALA as a predictive biomarker for future dementia. In the InCHIANTI study on 935 elderly subjects [68], using epidemiological analysis, the participants with dementia were found to have a significantly lower level of ALA (0.34% vs. 0.39%; *p* < 0.05) than the participants with normal cognitive function. This suggests that dementia is associated with low plasma ALA concentrations. Zhang [69] and Thomas [70] in their articles, were interested in the association between dementia and different dietary patterns, including the MIND Diet (Mediterranean DASH Intervention for Neurodegenerative Delay Diet), characterized by higher intakes of fish such as salmon, herring, tuna, and nuts, as well as fruits and vegetables, in middle-aged subjects with incident dementia and abnormal brain structures. The large population-based study by Zhang et al. [69], which included 114.684 adults over the age of 40 who followed different dietary patterns, found that about 0.42% of participants developed dementia. After adjusting for age and sex, it was found that subject who followed the MIND diet, which contains fish and nuts and therefore high levels of total PUFAs, may be associated with a low risk of dementia; furthermore, the same dietary patterns were positively associated with higher brain volumes in several regions, including the parietal and temporal lobes, hippocampus, and thalamus. This is important because volume loss, particularly in the temporal lobe and hippocampus, has been suggested to be a predictive biomarker of incident dementia [71]. Here, we presented a comprehensive picture of the consistent associations of dietary patterns with dementia risk and brain health, suggesting a potential role for diet in slowing the progression of brain atrophy. Similarly, Thomas et al. [70] investigated the association between the MIND diet, adapted to the French population, and grey matter volume, white matter microstructure, and incident dementia. This longitudinal study found that better adherence to the MIND diet was associated with lower diffusivity values in the splenium of the corpus callosum, which translates into a lower risk of developing dementia. The results of both studies therefore provide further evidence for the role of a diet rich in PUFAs in the prevention of dementia.

**Table 1 ijms-25-12479-t001:** Summary of studies including the type of studies, the enrolled population, the area of investigation, type of nutrients, and conclusions.

Study	Type of Study	Population	Area of Investigation	Evaluation/Supplement	Conclusions
Gustafsson et al., 2022 [31]	Prospective study	68 pregnant women Mean age: 30.49 y.o.	ADHD	Evaluated ALA AA EPA CLA Omega-6	Women with severe ADHD have: higher n-6 levels higher AA/EPA ratios lower EPA levels higher TNF-alpha levels
Pinar-Martì et al., 2023 [32]	Randomized intervention, two-arm, controlled trial	771 adolescents Mean age: 13.9 y.o.	ADHD	Supplemented with walnuts (30 g/day)	Walnuts (ALA) improved sustained attention, fluid intelligence, and ADHD symptoms.
Pinar-Martì et al., 2023 [33]	Randomized controlled trial	372 adolescents Mean age: 13.8 y.o.	ADHD	Evaluated ALA DHA	Higher blood levels of ALA appeared to result in lower impulsivity. DHA was associated with attention performance in typically developing adolescents.
Zielinska et al., 2019 [35]	Observational study	39 breastfeed infants and 39 women Mean age: 30.9 y.o. and 6.6 months	Psychomotor development	Evaluated ALA LA EPA DHA AA Carotenoids	The following were associated with increased psychomotor development: higher ALA levels higher DHA levels higher PUFA levels
Nomura et al., 2023 [36]	Two-sample MR analysis	72.517 participants	AN	Evaluated ALA LA AA EPA DHA DPA	PUFAs were not associated with AN.
Caspar-Bauguil et al., 2012 [37]	Observational study	22 women with AN	AN	Evaluated ALA LA Other PUFAs	Women with anorexia nervosa had lower PUFA levels no LA deficit no ALA deficit
Osuna et al., 2023 [49]	Observational case–control study	95 adolescents Mean age: 15 y.o.	Depression	Evaluated ALA EPA DHA	Adolescents with major depressive disorder had: lower EPA levels lower DHA levels lower PUFA levels lower AA levels higher n-6/n-3 ratios
Zeng et al., 2022 [50]	Two-sample MR analysis	500.199 healthy controls	Depression	Evaluated ALA EPA	A higher ALA level was associated with an increased risk of depression.
Chaves et al., 2022 [51]	Multicenter prospective cohort study	13.879 adults Mean age: 52 y.o.	Depression	Evaluated DHA ALA EPA	Higher ALA and higher EPA levels were associated with a decreased incidence of depressive episodes.
Fatemi et al., 2020 [52]	Cross-sectional study	300 women Mean age: 33.5 y.o.	Anxiety	Evaluated ALA Oleic acid PUFAs	Higher ALA, PUFA, and oleic acid levels were associated with a lower anxiety score in women.
Harauma et al., 2023 [2]	Double-blind, parallel, comparison study	250 primiparous, postpartum women	PPD	Supplemented with Perilla oil Fish oil 2.4 g/day of ALA 1.3 g/day of EPA 0.4 g/day of DHA	Higher ALA levels during pregnancy were associated with mental health postpartum.
Evans et al., 2012 [53]	Cross-sectional study	27 bipolar subjects	BD	Evaluated AA DHA EPA ALA Omega-6	The following were associated with increased suicidal history in BD: higher AA/DHA ratios higher AA/EPA ratios higher n-6/n-3 ratios higher DHA/ALA ratios higher DHA/EPA ratios lower EPA/ALA ratios lower AA levels
Gracious et al., 2010 [47]	Randomized, placebo-controlled trial	51 children/adolescents Mean age: 13 y.o.	BD	Supplemented with flaxseed oil (550 mg of ALA)	Higher AA and higher DPA levels in children with BD.
Current et al., 2023 [63]	Cross-sectional investigation trial	833 adults Mean age: 67.1 y.o.	Cognitive status	Evaluated ALA LA	The following were associated with better spatial memory and cognitive status: higher ALA levels lower LA levels lower n-6/n-3 levels
Dhillon et al., 2023 [64]	Observational study	78 South Australian adults Mean age: 75.75 y.o.	AD	Evaluated AA Stearic acid LA DHA GLA	Patients with AD had: higher stearic acid levels higher AA levels lower LA levels lower GLA levels lower ALA levels
Ogawa et al., 2023 [66]	Intervention study	60 women and men (adults) Mean age: 72 y.o.	Verbal fluency	Supplemented with flaxseed oil (3.7 g/day) - 2.2 g/day of ALA	A higher ALA level improved cognitive functions and verbal fluency.
Yamagishi et al., 2017 [67]	Intracohort case–control study	7586 Japanese adults from 0 to 74 y.o.	Dementia	Evaluated ALA PUFAs	ALA is considered a new biomarker for future dementia.
Cherubini et al., 2007 [68]	Epidemiological study	935 older people Mean age: 72 y.o.	Dementia	Evaluated FA ALA Palmitic acid	Patients with dementia have: higher palmitic acid levels lower ALA levels
Rog et al., 2020 [42]	Observational study	80 adults aged 18 to 65 y.o.	SCZ	Evaluated ALA AA DHA EPA LA	Lower omega-3 levels in SCZ.
Jones et al., 2021 [43]	Two-sample Mendelian randomized study	Adults with UHR	SCZ	Evaluated Omega-3 Omega-6	The following were associated with a decreased risk of SCZ: higher omega-3 levels higher omega-6 levels
Thomas et al., 2022 [70]	Longitudinal study	1412 French older adults Mean age: 75.8 y.o.	Dementia	Evaluated ALA EPA DHA	The MIND diet was associated with a lower dementia risk.
Zhang et al., 2023 [69]	Large-scale, population-based study	114.684 participants Mean age: 56.8 y.o.	Dementia	Evaluated ALA EPA DHA	The MIND diet containing ALA, EPA, and DHA was associated with a decreased risk of dementia.
Amminger et al., 2015 [40]	Double-blind, randomized, controlled-trial	81 UHR adolescents Mean age: 16.4 y.o.	Psychosis	Supplemented with marine oil fish (220 mg of PUFAs/day): ALA EPA DPA DHA LA AA NA	Higher ALA levels can be used as a marker for preventing psychotic disorder.

Abbreviations: UHR, ultra-high risk for psychosis; ADHD, attention-deficit/hyperactivity disorder; AN, anorexia nervosa; PPD, postpartum depression; BD, bipolar disorder; AD, Alzheimer’s disease; SCZ, schizophrenia; ALA, alpha-linolenic acid; EPA, eicosapentaenoic acid; DHA, docosahexaenoic acid; DPA, docosapentaenoic acid; LA, linoleic acid; AA, arachidonic acid; NA, nervonic acid; PUFAs, polyunsaturated fatty acids.

## 3. Discussion

The present narrative aimed to investigate the efficacy and benefits of PUFA administration in subjects with mood, behavioral, and neurodegeneration-related disorders in different populations, from children and adolescents to the elderly, with a particular focus on depressive disorders in pregnant women.

Specifically, the following areas were analyzed: neurodevelopmental disorders, eating disorders, psychosis, affective diseases, and cognitive impairment.

In the area of neurodevelopment, we focused on ADHD, autistic spectrum disorders, intellectual disabilities, and learning disabilities. We noted that the human studies showed promising conclusions: the consumption of both omega-3 and omega-6 was much lower in subjects presenting with ADHD symptoms [30]. A population of pregnant women was also considered, and it was found their EPA level and AA/EPA ratio were much lower than those of the controls [31]. The interventional studies by Pinar-Martì and Gignac [32] are interesting. They studied the benefits of giving walnuts (30 g/day), which are particularly rich in ALA, to a population of adolescents and observed a significant improvement in attention, fluid intelligence, and ADHD symptoms, suggesting an effect of this essential fatty acid on the central nervous system. The efficacy and validity of this innovative dietary strategy is supported by another interventional trial, which once again observed the usefulness of an ALA-rich diet in reducing impulsivity, a typical symptom of this neurodevelopmental disorder, and improving both attention and cognitive reaction time in a young population. These interventional studies show that, although studies are still rather scarce and there are many more animal studies than human ones, this is an area of research worth investigating, considering that neurodevelopmental disorders are often underestimated and the consequences they can have in the medium to long-term (e.g., global impairment of social skills and intelligence, and deficits in personal, social, academic, or occupational functioning) are ignored [72].

Regarding eating and nutritional disorders, we focused on anorexia nervosa and bulimia, disorders that are common in the younger population. The results for these diseases are particularly heterogeneous for several reasons, including the limited number of subjects recruited and the various study designs. No interventional trials in human cohorts were found and analyzed, only observational and retrospective studies in which the absence of ALA and LA, precursors of omega-3 and omega-6, was observed in the plasma membrane of the erythrocytes of the subjects that were recruited. These findings could be significant, indicating that a supplementation strategy based on essential fatty acids could probably be useful in the prevention of these eating disorders, but the data are too scarce and supplementation of subjects with high levels of these nutrients has not yet been proven to be effective. From this point of view, we are a long way from knowing whether PUFAs could be helpful in this type of eating disorder.

A very interesting aspect of mental health is the group of psychoses, which includes schizophrenia, schizoaffective disorder, and brief psychotic disorder. These disorders affect the general population, with no particular focus on one area. Different types of studies were analyzed for this area of research, including randomized and interventional studies with young subjects, such as 81 adolescents at high risk of psychosis (UHR). The analysis of the fatty acid composition of the plasma membrane of their erythrocytes revealed a significant ALA deficiency, suggesting that supplementation with this nutrient could prevent the onset of this disorder. Other observational studies and two-sample Mendelian randomization studies [43] in humans evaluated the effects of omega-3 supplementation on schizophrenia patients. The results confirmed a strong association between elevated omega-3 (EPA) levels and a reduced risk of developing schizophrenia, whereas the results regarding ALA levels were more heterogeneous and controversial; specifically, a positive association was found between elevated ALA levels and an increased risk of developing schizophrenia. Given the amount of heterogeneous data and particularly the inconsistent results, more research is needed to confirm that PUFA supplementation is effective in preventing the onset of schizophrenia and psychotic symptoms. Given the paucity of literature on ALA and more abundant literature on EPA/DHA, the number of interventional studies conducted with ALA-rich food supplementation would need to be increased to be able to state with certainty that this nutrient is indeed effective in reducing psychotic and schizophrenia-related symptoms.

Affective disorders include depression, the world’s leading cause of disability. Its onset is increasingly occurring early in childhood and adolescence, increasing the risk of developing in adults and the elderly. The number of studies considered was considerable and the type varied: there were randomized, placebo-controlled, observational, and interventional studies. In subjects with bipolar disorder, supplementation with flaxseed oil, which is particularly rich in ALA, was found to be helpful in reducing the symptoms associated with BD. The literature is also very heterogeneous on this point: although there are data clarifying the actual efficacy of EPA, especially ALA, in reducing depressive and borderline symptoms, some case–control studies and two-sample analyses have suggested that although EPA is associated with a protective effect against the risk of depression, the same cannot be said for ALA, which has been suggested to be a potential risk factor. This association was also studied in a population of pregnant women at high risk of developing postpartum depression. A significant reduction in the frequency of episodes of postpartum depression was observed after 12 weeks of supplementation with ALA-rich fish oil and perilla, suggesting that this could be used as an adjunctive therapy to alleviate depressive symptoms in individuals who may be more susceptible to developing the condition. The heterogeneity of data from studies in scientific literature is probably due to the specific role of PUFAs in the treatment of disorders such as depression and DB still being unclear. Currently, it cannot be confirmed that PUFAs are effective in treating the symptoms associated with mood and behavioral disorders. Further research is needed to differentiate between depression and DB and to increase the number of studies that can effectively test diets containing nuts, flaxseed, or hempseed, which are known to be rich in ALA, in this population.

When we talk about cognitive impairment, we mainly mean dementia, for which there are mainly pharmacological therapies. Alternatively, there are studies confirming the efficacy of the Mediterranean diet, which is particularly rich in PUFAs, fiber, and polyphenols, in preventing cognitive decline. In this field, cross-sectional studies were performed, sometimes using animal models, to test the relationship between the consumption of ALA-rich nuts and improvements in spatial memory. The explanation for the efficacy of this diet may lie in the fact that PUFAs, especially ALA, EPA, and DHA, enhance neurogenesis in the hippocampus of adult subjects by promoting synaptic plasticity, modulating the expression of synaptic proteins and, above all, increasing the serum levels of neurodegeneration-related peptide hormones such as BDNF and IGF-1. Reduced levels of these peptide hormones increase neuronal atrophy, thereby increasing the risk of neurodegenerative disease and decreasing the level of neuroprotection. However, it is still too early to define the overall efficacy of this fatty acid on the human population suffering from neurodegenerative diseases. Promising results have come from interventional studies [66]. For 12 weeks, 60 elderly people were given nuts containing a total of 2.2 g/day of ALA. Improvements in verbal fluency were observed in the patient group compared with the control group. Similar results were also observed in the studies by Zhang et al. [69] and Thomas et al. [70], in which it was observed that adherence to a specific diet, a combination of the DASH diet and the Mediterranean diet, which is high in PUFAs, was directly associated with a very low risk of developing dementia. This aspect is worth highlighting as it underlines the effectiveness of PUFAs in slowing the progression of brain atrophy. The results of the studies focusing on dementia are very promising. However, there is a need to carry out case series and to focus more on interventional studies rather than retrospective and observational studies.

In addition, regarding clinical trials focusing on ALA, it is also important to note that the exact amount of ALA that is converted into EPA and DHA is still somewhat unknown. Plant organisms, for example, phytoplankton, can convert ALA into its two higher derivatives and the same applies to humans, but with lower yields. According to the AFFSA (French Food Safety Agency), due to the low yields of the endogenous biosynthetic pathway, an average bioequivalence factor of 10 has been used for the conversion of EPA and DHA to ALA. In other words, 1 g of ALA leads to the formation of 100 mg of EPA and DHA. This means that, with this conversion rate, the amount of ALA that needs to be administered daily for this fatty acid to be converted into its metabolites is extremely high.

The main problem with scientific literature is that the studies are too heterogeneous in terms of the nutrients administered and daily dosages, ranging from 30 g of walnuts per day [32] to 550 mg of ALA [47]. According to the LARNs, the daily requirements for PUFAs are as follows: 2.5/2.6 g of ALA per day, and from 2 to 4 g of EPA and DHA per day. The doses administered in clinical trials vary considerably. This methodological problem makes it difficult to compare data. One aspect that could be improved would be to encourage interventional studies in which foods rich in ALA from plant sources are fed in equal amounts and compared; this same protocol should be used in interventional studies evaluating the feeding of food of animal origin that is rich in EPA and DHA. One factor that affects the evaluation of the results is the duration of the study and was similar between studies with most studies evaluating a 3-month period (approx. 12 weeks).

Despite the promising results and proven efficacy of daily nuts supplementation, as seen in some of the interventional studies mentioned above, the effects of this supplementation in the areas of neurodevelopment, eating disorders, cognitive decline, and affective disorders are still very much unexplored and it may be premature to state with certainty that PUFAs could represent an innovative new integrative strategy for these types of disorders. For example, it may be useful to combine supplementation of these nutrients with conventional drug therapy to minimize the risk of cognitive decline and limit long-term neuronal damage. Many more studies still need to be conducted before significant results can be reported in scientific literature.

## 4. Materials and Methods

This narrative review was performed using the PubMed, Google Scholar, and Scopus databases, searching for all studies published in English from 2005 to 2023. The most used keywords were (alpha-linolenic acid) AND (psychiatry OR mental health OR schizophrenia OR depression OR bipolar disorder OR anorexia OR psychosis OR neurodevelopment OR dementia OR Alzheimer’s disease OR adult ADHD). The inclusion criteria were studies in humans; systematic reviews; Mendelian randomization studies; case–control studies; cross-sectional studies; studies using dietary supplementation such as supplementation with PUFAs, specifically ALA; epidemiological studies; observational studies; and studies carried out on children, adolescents, and middle-aged and older adults, pregnant and lactating women, and elderly people. Studies on other effects of ALA (e.g., on metabolic syndrome, hypercholesterolemia, and hypertriglyceridemia) and studies on the relationship between mental health and PUFAs/ALA published before 2005 were not included.

## 5. Conclusions

In the present review, we looked at how PUFAs, particularly ALA, EPA, and DHA, can be helpful and supportive in the prevention of behavioral, mood, and neurodegenerative diseases. In most of the studies included, from development to old age, it was observed that the levels of PUFAs were particularly low in subjects presenting with these conditions, which suggests that supplementation with polyunsaturated fatty acids may be beneficial in preventing or treating these conditions, given their known neuroprotective role. However, due to the paucity of human studies in this area, further research is needed to confirm the efficacy and usefulness of this supplementation strategy in the context of neurological and mental illnesses.

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
