# Peer review of "The Role of Alpha-Linolenic Acid and Other Polyunsaturated Fatty Acids in Mental Health: A Narrative Review"

_ijms, 2024, doi:10.3390/ijms252212479_

Round 1
Reviewer 1 Report
Comments and Suggestions for Authors
This narrative review merely updates the potential role of omegas in mental health. However, I believe that readers would benefit if the authors included a propositions section and an evaluation of the cited articles. Other studies show that omegas do not play a relevant role in mental health. Therefore, making this counterpoint is essential for there to be, in fact, a contribution to this area of ​​knowledge.

Reviewer 2 Report
Comments and Suggestions for Authors
The present narrative review by Bertoni et al. focuses on a very interesting topic of research with the potential for relevant health outcomes both from a scientific and from a societal perspective. Notwithstanding, I found this research work rather underdeveloped. The body of literature the authors summarised is not accurately presented and, more often than not, strong conclusions are drawn either from findings on animal studies, or without providing adequate information on the methodologies employed and the corresponding results. The Discussion is particularly scant with no mention of important limitations to keep in mind when considering available research studies in this area.
Despite finding the research topic of this review extremely fascinating, I have several major concerns about its conduct. Regretfully, I do not believe that this work is worthy of publication as it stands. In the attached document, I have listed and detailed my main sources of concern for the authors to reflect on and address, should they decide to amend their manuscript and resubmit.

Please check for spelling, grammar and punctuation throughout the text. I ave found quite a few typos and subject-verb disagreements.
Round 2
Reviewer 1 Report
Comments and Suggestions for Authors
I thank the authors for taking my suggestions into consideration.
Author Response
Thank you for your comments.
Reviewer 2 Report
Comments and Suggestions for Authors
Please find my comments in the attached document.

The use of the English language can be improved. There are several grammatical errors that must be addressed to improve the clarity of the manuscript.
